# Central Venous Catheter Cannulation in Pediatric Anesthesia and Intensive Care: A Prospective Observational Trial

**DOI:** 10.3390/children9111611

**Published:** 2022-10-23

**Authors:** Václav Vafek, Tamara Skříšovská, Martina Kosinová, Eva Klabusayová, Tereza Musilová, Tereza Kramplová, Jana Djakow, Jozef Klučka, Jiří Kalina, Petr Štourač

**Affiliations:** 1Department of Paediatric Anaesthesiology and Intensive Care Medicine, University Hospital Brno and Faculty of Medicine, Masaryk University, Jihlavská 20, 62500 Brno, Czech Republic; 2Department of Simulation Medicine, Faculty of Medicine, Masaryk University, Kamenice 5, 62500 Brno, Czech Republic; 3Paediatric Intensive Care Unit, NH Hospital Inc., 26801 Hořovice, Czech Republic; 4RECETOX, Faculty of Science, Masaryk University, Kotlářská 2, 61137 Brno, Czech Republic

**Keywords:** ultrasound, central venous catheter, cannulation, pediatric, intensive care

## Abstract

Currently, ultrasound-guided central venous catheter (CVC) insertion is recommended in pediatric patients. However, the clinical practice may vary. The primary aim of this study was the overall success rate and the first attempt success rate in ultrasound-guided CVC insertion versus anatomic-based CVC insertion in pediatric patients. The secondary aim was the incidence of associated complications and the procedural time. The physician could freely choose the cannulation method and venous approach. Data were collected for 10 months. Overall, 179 patients were assessed for eligibility and 107 patients were included. In almost half of the patients (48.6%), the percutaneous puncture was performed by real-time ultrasound navigation. In 51.4% of the patients, the puncture was performed by the landmark method. The overall success rate was 100% (n = 52) in the real-time ultrasound navigation group, 96.4% (n = 53) in the landmark insertion group, (*p* = 0.496). The first percutaneous puncture success rate was 57.7% (n = 30) in the real-time ultrasound navigation group and 45.5% (n = 25) in the landmark insertion group, (*p* = 0.460). The data show a higher overall success rate and the first success rate in the US-guided CVC insertion group, but the difference was not statistically significant.

## 1. Introduction

Securing the intravenous line is one of the fundamental interventions in pediatric anesthesia and intensive care. Central venous catheters (CVC) are mostly indicated for long-term stays in an intensive care unit (ICU), in cases of hemodynamic instability and the need for vasopressor therapy, for hypertonic solutions administration, or for parenteral nutrition. The predominant method of CVC insertion is the Seldinger technique (over the wire). There are currently two options to cannulate the central venous system: the landmark technique and the real-time ultrasound-guided (US-guided) technique of CVC cannulation. Current evidence suggests that the real-time US-guided technique is an effective and safe method for CVC cannulation in adults and children [1,2,3]. Despite many recommendations, the traditional technique, based on the identification of anatomic landmarks, is still widely used [1,4]. Data from metanalyses [1,5,6] show the superiority of the real-time US-guided CVC technique over the landmark method. US guidance for CVC cannulation reduces the time of cannulation [7], significantly increases the success rate [1,4,5,8,9], reduces inadvertent arterial punctures [1,6,7,10], and decreases the number of attempts to gain placement [4,8,9,10], as well as the number of complications during central venous cannulation [8,9,11]. Common sites for CVC access are the internal jugular (IJ), subclavian (SCV), and femoral veins (FV). In the pediatric population, supraclavicular US-guided access to the subclavian or the brachiocephalic vein (BCV) appears to be the safe alternative to IJV and FV [8,12,13,14]. In comparison to the infraclavicular approach, the US-guided supraclavicular approach allows in-plane puncture to display the progression of the whole needle, including the tip. Therefore, this method is associated with a lower risk of pneumothorax and artificial artery puncture. It also relates to shorter puncture time and decreased guidewire misplacement [13]. The recent *EJA* (*European Journal of Anaesthesiology*) guidelines recommended the US-guided method as the preferred one for the cannulation of IJV, FV, and BCV [3]. Complications associated with CVC cannulation are commonly divided into mechanical, infectious, and thromboembolic complications. Mechanical complications may include pneumothorax, artificial arterial punctures, hematomas, pulmonary embolism, nervous system injury, and, rarely, death [1,11,15]. The incidence of mechanical complications is directly associated with the number of cannulation attempts [16]. The aim of this study was to compare the efficacy and safety of the US-guided and landmark techniques of routine CVC insertion in pediatric patients in the pediatric ICU and in the operating room. The hypothesis of the trial was the higher overall success rate and higher first attempt success rate in the US-guided group. The primary aim of the study was the overall success rate and the first success rate in US-guided CVC insertion and in the landmark CVC insertion in pediatric patients. The secondary aim was the incidence of associated complications and the time from the first percutaneous puncture to the definite CVC securing in place. 

## 2. Materials and Methods

This prospective observational cohort trial was conducted from March 2020 to December 2020 at the Tertiary Pediatric Anesthesia Center: Department of Pediatric Anaesthesiology and Intensive Care Medicine, University Hospital Brno and Faculty of Medicine, Masaryk University, Czech Republic. The trial was approved by the Ethics Committee of the University Hospital Brno, Jihlavská 20, 625 00, Brno, Czech Republic (Approval Number: 02/2020, Chairperson: Pharm. Dr. Kozáková, date of approval: 1 February 2020), and the trial was registered on clinicaltrials.gov (Clinicaltrials.gov identifier: NCT04211116). Informed consent (other than the consent for the CVC cannulation) was not required. All CVC insertions performed in the anesthesiology and PICU division of the Department of Pediatric Anaesthesiology and Intensive Care Medicine were eligible for inclusion. All operators were trained in basic ultrasound skills prior to the trial initiation. According to the trial design, the choice of the CVC insertion method and the site of the CVC insertion were based on operators’ decision only. Two possible ultrasound machines were available: SonoSite EDGE^®^ ultrasound with a linear probe (5–10 MHz) and hockey-stick probe (6–13 MHz) and GE Healthcare^®^Logiq P5 ultrasound with a linear probe (3.4–10.8 MHz). Pediatric patients (up to 19 years of age) in whom the attending physician indicated CVC insertion under general anesthesia were included in the trial. The exclusion criterion was only an age outside the age limit. All CVCs were inserted under general anesthesia or analgosedation. According to the study design, the CVC insertion method and the site were the operators’ choice. The following data were collected: demographic data, indication for CVC insertion, primary planned site for insertion, implementation of ultrasound, eventual reason for non-use of ultrasound, site of actual CVC cannulation, number of attempts, technical parameters of central venous catheter (diameter in French, number of lumens, depth of the insertion) and complications during cannulation (artificial punction of artery, number of attempts to insert the guidewire). After successful cannulation, a chest X-ray was taken at the discretion of the attending physician. Data from the control X-ray (malposition of catheter and occurrence of pneumothorax) were analyzed. For the data analysis, standard descriptive methods such as mean, standard deviation (SD), median, and range (minimum, maximum) were used. Categorical (qualitative) parameters were summarized using absolute and relative frequencies. Differences in all monitored parameters related to the patient and the introduction of CVC according to the method of execution (US-guided method vs. landmark method) were evaluated using the Mann–Whitney test (continuous parameters) or Fisher’s exact test (categorical parameters).

## 3. Results

The patients were recruited from 1 March to 31 December 2020. In total, 179 patients were assessed for eligibility; 72 patients were not included due to missing or incomplete data and 107 patients were included and analyzed (Figure 1). 

The landmark method was used in 51.4% of patients (n = 55). The ultrasound-guided method was used in 48.6% patients (n = 51). In the US-guided group, older patients were enrolled (in mean older by about 2.5 years, median 6.7 years) (Table 1).

The indication of the CVC insertion, punction, and type of anesthesia was comparable between the two groups. The differences between data were not statistically significant (*p* = 0.909). Statistically significant difference was described in the primary planned site for insertion. In the landmark group, the left subclavian vein was selected in 90.9% for cannulation (n = 50), followed by the right subclavian vein in 9.1% (n = 5). Other sites have not been used. In the US-guided group, the most common site for cannulation was the left brachiocephalic vein, which was selected in 55.8% of patients (n = 29), then the right jugular vein in 19.2% (n = 10), and the right brachiocephalic vein in 13.5% (n = 7), (*p* < 0,001) (Table 2).

The overall success rate was 100% (n = 52) in the real-time ultrasound navigation group and 96.4% (n = 53) in the landmark insertion group, without statistical significance (*p* = 0.496). The number of required percutaneous attempts (number of necessary skin punctures) was comparable between the two groups (*p* = 0.460). There was a statistically significant difference in the final planned site for insertion. The left subclavian vein was cannulated in 89.1% (n = 49) in the landmark group, followed by the right subclavian vein in 9.1% (n = 5). In the US-guided group, the most common site for cannulation was the left brachiocephalic vein, cannulated in 50.0% patients (n = 26), followed by the right jugular vein in 15.4% (n = 8) and the right brachiocephalic vein in 15.4% (n = 8) (*p* < 0.001). A statistically significant difference was described in the number of operators during cannulation. In the US-guided group, two operators were present during the cannulation in 46.2% (n = 24), compared to the 27.3% (n = 15) in the landmark group (*p* = 0.047). Additionally, the difference in the size of the catheter was statistically significant (*p* = 0.042) (Table 3). 

The level of experience (years of practice) does not affect the first puncture success rate (*p* = 0.25) or the overall success rate (*p* > 0.5). The time of the insertion tended to be longer in the US-guided group, but the difference did not reach statistical significance (19.3 ± 11.9 s vs. 14.5 ± 7.9 s) in the landmark group (*p* = 0.059) (Table 4).

The overall incidence of complications was comparable between the two groups. The incidence of artificial arterial puncture was 7.7% (n = 4) in the US-guided group vs. 3.6% (n = 2) in the landmark group (*p* = 0.429). The insertion of the guidewire on the first attempt was achieved in 69.2% (n = 36) in the US-guided group vs. 87.3% (n = 48) in the landmark group (*p* = 0.075). The control chest X-ray was taken in 94.2% of patients (n = 49) in the US-guided group vs. 94.5% (n = 52) in the landmark group (*p* = 1.000). Malposition of the catheter was described in 15.4% (n = 8) in the US-guided group vs. 9.1% (n = 5) in the landmark group (*p* = 0.222). Pneumothorax was described in 1.9% (n = 1) in the US-guided group vs. 1.8% (n = 1) in the landmark group (*p* = 0.125) (Table 5).

## 4. Discussion

Central venous cannulation remains one of the most prevalent procedures in pediatric intensive care. Besides the common indications derived from adult intensive care (inotropic and vasopressor therapy, intravenous cardiac stimulation, parenteral nutrition, hypertonic solution administration, and invasive hemodynamic monitoring), the difficulties with peripheral vein cannulation and the frequent need for blood sample analysis in infants and neonates make central venous catheters even most prevalent in PICUs and NICUs compared to adult ICUs. Currently, jugular, subclavian, and femoral veins are the most preferred sites for CVC insertion. Historically, anatomical-based CVC insertion was preferred. However, in the last two decades, ultrasound vein visualization and real-time puncture were found to be more effective and safer. The anatomical differences in infants and neonates (relatively big head and short neck, and decreased blood flow to the lower part of the body) could have a negative impact on the effectiveness of jugular and femoral vein cannulation. The subclavian vein therefore seems to be the ideal option. However, serious risks of pneumothorax and hemothorax are associated with blind anatomical infraclavicular or supraclavicular puncture. Based on many studies and meta-analyses [1,3,4,5,7,8,10,11], US-guided CVC cannulation in children is the preferred method for the jugular and femoral veins, mainly due to higher safety and efficacy in children. Due to difficult subclavian vein visualization from the infraclavicular ultrasound window, the USG real-time puncture could be used by experienced operators. However, the optimal ultrasound view of the subclavian vein could be reached from the supraclavicular ultrasound view together with the possibility of in-plane real-time vein cannulation (long axis view—the whole length of the needle is displayed on the USG image). These factors could significantly increase the efficiency and safety of CVC insertion. Nevertheless, the landmark method is still widely used, probably even more in children than in adults. This may be associated with potentially lower CVC cannulation safety. Based on these results and experience with using US-guided cannulation, we expected to confirm the hypothesis, i.e., the higher overall success rate and higher first attempt success rate in the US-guided group. The technique of the ultrasound-guided puncture CVC is based on the same principles as in adults. A basic US machine should be sufficient for a puncture; a linear probe is the most suitable (limited tissue penetration, superior tissue visualization due to the higher frequency). In small children (infants and neonates), the use of a hockey-stick linear probe could be required due to the anatomical proportions [1,3,4,5,7,8,10,12,13,14,17,18]. 

It is recommended to perform a real-time USG vein examination to measure the vein size, choose the appropriate catheter size, and find the most suitable place for the puncture before preparing the sterile field for the cannulation. After the USG real-time puncture, the ultrasound should be used for the verification of the guidewire and catheter position. In the pediatric population, several approaches can be used for CVC cannulation. The internal jugular vein is the most accessible, real-time USG technique recommended for cannulation. The risks include easy compressibility, a tendency to collapse, and a higher risk of artificial carotid artery puncture in comparison to other sites. The subclavian vein does not collapse even in children in shock or with hypovolemia. Real-time ultrasound or ultrasound-assisted puncture is recommended for SCV cannulation. The brachiocephalic vein is accessible from the supraclavicular approach; the in-plane real-time USG cannulation is the recommended approach. Femoral vein cannulation is recommended in real-time imaging. However, a higher risk of catheter infections is associated with VF cannulation [3,10]. 

Supraclavicular access to the brachiocephalic vein in children can be a very effective cannulation technique, provided the operator is trained in the USG cannulation technique. In comparison to the infraclavicular approach, the USG supraclavicular approach allows in-plane cannulation, displaying the progress of the whole needle, including the tip. In the youngest pediatric patients, ultrasound imaging of the venous system may be insufficient, and cannulation can be technically very difficult. The supraclavicular approach became more and more popular in our department, especially in the youngest pediatric patients. It is possible to visualize the venous system and perform a real-time cannulation even in newborns weighing less than 1500 g. The use of a hockey-stick probe is essential in this age category due to the neonate-specific anatomical differences (large head, short neck). Previously published data also confirms that the supraclavicular US-guided approach is probably the easiest and safest option for CVC cannulation in neonates and small infants [8,17,18,19,20]. When analyzing the primary outcome of this trial, the overall success rate of 100% (n = 52) versus a 96.4% (n = 53) success rate was described in the USG vs. landmark method, (*p* = 0.496). These results are far higher compared to previously published data [1,5,21,22]. Another part of the primary outcome was the first attempt’s success rate of cannulation. In our cohort, the majority of cannulations were limited by first and second attempts due to successful cannulation with a higher success rate of the first attempt in the USG group: 57.7% (n = 30) vs. 45.5% (n = 25), however without statistical significance, (*p* = 0.460). These results are comparable to the previously published data [5,7,17,18,19]. The incidence of associated complications: arterial puncture (7.7%, n = 4 USG versus 3.6%, n = 2, landmark, *p* = 0.429) and pneumothorax (1.9%, n = 1, USG versus 3.6%, n = 1, landmark, *p* = 0.125) were comparable between the groups.

Although it is well known that ultrasound technique mastering requires training, the operators’ experience defined by years of practice was not associated with a higher overall success rate (*p* > 0.5) or the first puncture success rate (*p* = 0.25). There can be several potential limitations of the study.

Our study has several limitations. First, the trial was based on an observational study design. The other limitation is the low number of patients in our experiment and a relatively large group of excluded patients. Despite the fact that all operators were conventionally trained with USG and USG cannulation, significant differences could be based on each operator’s cannulation method preference. On the other hand, the aim of the study was to evaluate the USG cannulation success rate versus the landmark method in the mixed PICU staff in actual clinical settings (staff with mixed USG cannulation skills) to reach higher possible external validity of the results. Although this parameter is not significant, we can see a trend that senior operators tended to choose the landmark method (*p* = 0.48). The relationship between these parameters is not significant, probably due to the low number of patients.

Despite our results, previously published data, including the metanalyses, favor the USG CVC cannulation. The most effective method seems to be cannulation of the brachiocephalic vein from the supraclavicular approach, which is also effective in infants and low-birth-weight infants.

## 5. Conclusions

Our data show a higher overall success rate and the first success rate in the US-guided CVC insertion group, but the difference did not reach statistical significance.

## Figures and Tables

**Figure 1 children-09-01611-f001:**
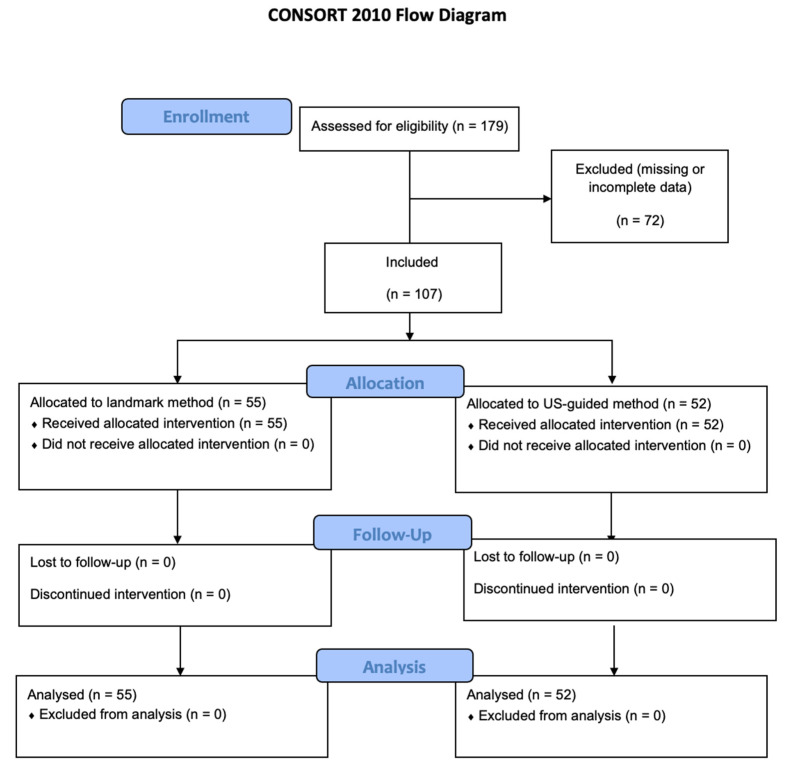
Flow diagram.

**Table 1 children-09-01611-t001:** Demographics.

	US-Guided (n = 52)		Landmark (n = 55)	
	Mean (SD)	Median (Min–Max)	Mean (SD)	Median (Min–Max)
Age (years)	6.6	7.1	4.1	0.3
	(5.9)	(0.0–17.7)	(5.9)	(0.0–17.5)
Height (cm)	108	121	86	60
	(48)	(34–186)	(48)	(33–180)
Weight (kg)	26	22	18	5
	(22)	(1–82)	(22)	(1–71)

**Table 2 children-09-01611-t002:** Indication.

	Category	US-Guided (n = 52)	Landmark (n = 55)	*p*-Value
		n (%)	n (%)	
Indication	Long-term I.V. access Parenteral nutrition Catecholamine’s Infusion therapy Antibiotics	28 (53.8%) 20 (38.5%) 2 (3.8%) 2 (3.8%) 0 (0.0%)	27 (49.1%) 24 (43.6%) 2 (3.6%) 1 (1.8%) 1 (1.8%)	0.909
Planned insertion site	Vena subclavia l. sin. Vena subclavia l. dx. Vena brachiocephalica l. sin. Vena brachiocephalica l. dx. Vena jugularis interna l. sin. Vena jugularis interna l. dx. Vena femoralis l. sin. Vena femoralis l. dx.	2 (3.8%) 0 (0.0%) 29 (55.8%) 7 (13.5%) 2 (3.8%) 10 (19.2%) 1 (1.9%) 1 (1.9%)	50 (90.9%) 5 (9.1%) 0 (0.0%) 0 (0.0%) 0 (0.0%) 0 (0.0%) 0 (0.0%) 0 (0.0%)	<0.001
Type of anesthesia	General anesthesia	51 (98.1%)	53 (96.4%)	1.000
	Sedation	1 (1.9%)	2 (3.6%)	

**Table 3 children-09-01611-t003:** Technical parameters of the punction.

	Category	US-Guided (n = 52)	Landmark (n = 55)	*p*-Value
		n (%)	n (%)	
Success cannulation	Yes No	52 (100%) 0 (0.00%)	53 (96.4%) 2 (3.6%)	0.496
Number of necessary attempts (necessary skin puncture)	1 2 3 4 More than 5	30 (57.7%) 11 (21.2%) 4 (7.7%) 2 (3.8%) 0 (0.0%)	25 (45.5%) 12 (21.8%) 3 (5.5%) 6 (10.9%) 9 (16.4%)	0.460
Final insertion site	Vena subclavia l. sin. Vena subclavia l. dx. Vena brachiocephalica l. sin. Vena brachiocephalica l. dx. Vena jugularis interna l. sin. Vena jugularis interna l. dx. Vena femoralis l. sin. Vena femoralis l. dx.	9 (17.3%) 2 (3.8%) 26 (50.0%) 3 (5.8%) 1 (1.9%) 8 (15.4%) 1 (1.9%) 1 (1.9%)	49 (89.1%) 5 (9.1%) 1 (1.8%) 0 (0.0%) 0 (0.0%) 0 (0.0%) 0 (0.0%) 0 (0.0%)	<0.001
Catheter size (Fr)	1.0 3.0 4.0 5.0 5.5 7.0 9.0 12.0	1 (1.9%) 4 (7.7%) 11 (21.2%) 9 (17.3%) 4 (7.7%) 21 (40.4%) 1 (1.9%) 1 (1.9%)	0 (0.0%) 15 (27.3%) 16 (29.1%) 8 (14.5%) 4 (7.3%) 12 (21.8%) 0 (0.0%) 0 (0.0%)	0.042
No of lumens	2	32 (61.5%)	43 (78.2%)	0.090
	3 or more	20 (38.5%)	12 (21.8%)	
No of operators	1 2	28 (53.8%) 24 (46.2%)	40 (72.7%) 15 (27.3%)	0.047
Years of practice	0–5 5–10 10 and more	28 (53.8%) 12 (23.1%) 12 (23.1%)	31 (56.4%) 4 (7.3%) 20 (36.3%)	0.048

**Table 4 children-09-01611-t004:** Technical parameters of the punction.

	US-Guided (n = 52)		Landmark (n = 55)		*p*-value
	Mean (SD)	Median (Min-Max)	Mean (SD)	Median (Min-Max)	
Depth of insertion (cm)	10.6	10.5	9.2	8.0	0.031
	(3.6)	(5.0–18.0)	(3.6)	(5.0–17.0)	
Time of insertion (min)	19.3 (11.9)	15,1 (2.0–50.0)	14.5 (7.9)	15.0 (2.7–42.0)	0.085

**Table 5 children-09-01611-t005:** Complications.

	Category	US-Guided (n = 52)	Landmark (n = 55)	*p*-Value
		n (%)	n (%)	
Artificial punction of artery	No Yes	48 (92.3%) 4 (7.7%)	53 (96.4%) 2 (3.6%)	0.429
Insertion of guidewire (No. of attempts)	1st 2nd 3rd or more	36 (69.2%) 11 (21.2%) 5 (9.6%)	48 (87.3%) 5 (9.1%) 2 (3.6%)	0.075
Periprocedural X-ray control	No	3 (5.8%)	3 (5.5%)	1.000
	Yes	49 (94.2%)	52 (94.5%)	
X-ray control and possible malposition	No Yes–no malposition Yes–malposition	7 (13.5%) 37 (71.2%) 8 (15.4%)	3 (5.5%) 47 (85.5%) 5 (9.1%)	0.222
Pneumothorax	No control Control–PNO Control–no PNO	7 (13.5%) 44 (84.6%) 1 (1.9%)	2 (3.6%) 52 (94.5%) 1 (1.8%)	0.125

## Data Availability

Data supporting reported results can be provided by the contact person: Jozef Klucka, Klucka.Jozef@fnbrno.cz; Tel.: +420-53223-4696.

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
