# Peer review of "Central Venous Catheter Cannulation in Pediatric Anesthesia and Intensive Care: A Prospective Observational Trial"

_children, 2022, doi:10.3390/children9111611_

Round 1

Reviewer 1 Report

The observational study “Central Venous Catheters Cannulation in Pediatric Anaesthesia and Intensive Care: Prospective Observational Trial” by Václav Vafek et al. described the differences of success rates between US-guided CVC insertion group and landmark insertion group. The conclusion showed that there was no statistical significance between these 2 groups.

In the abstract, the authors stated that the study was conducted over a 9-month period. However, there were 10 months from March 1st to December 31st 2020.

Of note, the difference of median age between those 2 groups was up to 2.5 years (older age in US-guided intervention group). The age and body weight of the patients could affect the success rate of CVC insertion, whether it was US-guided. So, the age of the cases between 2 groups should be similar.

The experience of the operator could be the most important factor on the success rates. However, there was no comparison in how experienced the operators are between these 2 groups. This could be the major bias in this study.

The case number is too low. More cases are needed for analysis.

Reviewer 2 Report

I am a neonatologist working in the NICU for 11 years, in a tertiary regional center. From daily practice, I must tell you that there is no need for ultrasound guidance when inserting a percutaneous CVC for neonatal patients. We only do a radiological examination after installing the CVC. It would be very difficult to use ultrasound guidance on a neonate that is around 600 grams. I think you should talk to the neonatologists in the NIC and create a special paragraph about percutaneous CVC in preterm babies.

Also, the conclusions are too short for such an article. You must add more details about the results of your research. Conclusions should be at least 10 rows ....in my opinion.

Round 2

Reviewer 2 Report

I am glad that you updated the article.